# Mind the ramp: Association between early life ramp use and spatial cognition in laying hen pullets

Alex Johny[1,2]*, Andrew M. Janczak[3], Janicke Nordgreen[4], Michael J. Toscano[1], Ariane Stratmann[1]¤

1 VPHI Institute, Centre for Proper Housing of Poultry and Rabbits, University of Bern, Zollikofen, Switzerland, 2 Graduate school of Cellular and Biomedical Sciences, University of Bern, Bern, Switzerland, 3 Faculty of Veterinary Medicine, Department of Production Animal Clinical Science, Norwegian University of Life Sciences (NMBU), Ås, Norway, 4 Faculty of Veterinary Medicine, Department of Paraclinical Sciences, Norwegian University of Life Sciences, Ås, Norway

¤ Current address: Centre for Proper Housing of Ruminants and Pigs, Federal Food Safety and Veterinary Office, Agroscope, Posieux, Switzerland
* alex.johny@unibe.ch

**Data Availability Statement:** The data and code for all analysis can be found at https://osf.io/9j2g8

**Funding:** AJ: The work was supported by the European Union's Horizon 2020 research and

## Abstract

Ramps facilitate earlier access to complex environments and increase early life voluntary exercise, which may positively affect the cognitive development of chickens. This study focused on quantifying individual differences in ramp use and its impact on spatial cognition of laying hen pullets. Sixteen identical pens were housed with Lohmann Selected Leghorn (LSL) chicks of which eight chicks from each pen were colour marked from one day of age (DoA) to serve as focal birds. We quantified overall ramp use (walk/run, wing-assisted incline running, and jump/fly to and from ramps) by scan sampling recorded videos for 6, 10, 12, 20, 27, 41, and 55 DoA for all focal birds. From 56 to 95 DoA, long and short-term spatial memory of three focal birds per pen were assessed in a holeboard test in three consecutive phases: cued, uncued and reversal. Mixed model analysis showed that the spatial cognitive abilities of the birds were linked to differences in ramp use frequency averaged across all observation days. Birds with higher ramp use made fewer reference (Estimate ± Confidence Interval = 0.94 [0.88, 0.99], p = 0.08) and working memory errors (Est ± CI = 0.77 [0.59, 1.00], p = 0.06) in the cued phase than birds with lower ramp use. In contrast, birds with higher ramp use made more reference memory errors (Est ± CI = 1.10 [1.01, 1.20], p = 0.05) in the reversal phase. Birds with higher ramp use also made more reference memory errors compared to birds with lower ramp use as the phases changed from cued to uncued (p = 0.001). Our results indicate that there might be a relationship between early life ramp use and spatial cognition of laying hens.

innovation programme under the Marie Skłodowska-Curie grant agreement No 812777. The funders had no role in study design, data collection and analysis, decision to publish, or preparation of the manuscript.

**Competing interests:** The authors have declared that no competing interests exist.

## Introduction

Spatial cognition is the ability of animals to acquire, use, recall and revise information concerning the spatial layout of their environment [1]. It is critical for an animal's survival, as it enables navigation through the environment, locating resources such as food, water, shelter, conspecifics, and avoiding predators. Early life experiences play a considerable role in the development of spatial cognitive abilities of animals [2,3] and the complexity of the habitat in this period is a defining feature. Rearing animals in complex environments has been shown to improve spatial learning and memory [4,5], as well as having positive developmental effects on the brain regions involved in spatial cognition in a range of taxa (rat: [6], atlantic salmon: [7], mice: [8]).

The domestic hen is an excellent model species to study the development of spatial cognitive abilities due to their precocial nature [9]. Precocial birds can be tested for cognitive functions from a very early age as they hatch with a fully developed sensory-motor system and rapid learning mechanisms [10]. Hens are able to navigate complex environments using landmarks [11], egocentric information [12] as well as topographical features of the environment [13]. They are also able to use multiple landmarks for orientation, possibly by forming cognitive maps [14]. As in several other species, the development of spatial cognition in domestic chicks has also been shown to be influenced by early life environment. Chicks reared in complex environments by either equipping the housing system with elevated perches [15] or ground level visual barriers [16] have been shown to improve their spatial skills in navigating vertical spaces later in life [15] and their egocentric orientation [16]. The visual barriers also induced changes in morphology of the hippocampus [17], the brain region involved in regulation of spatial navigation and memory [9].

Spatial cognition of domestic hens has been of scientific interest in applied research recently [18,19], owing to the changes in housing conditions over the last three decades [20]. An increasing number of laying hens are being housed in cage-free systems such as aviaries [21], which are structurally complex compared to cage systems (both battery and enriched) due to the vertically stacked tiers, within which essential resources such as perches, litter, food and water are located. Rearing birds in complex aviaries has been shown to benefit the spatial cognition [22] and working memory [23] of laying hens. However, the access to the three-dimensional environment is compromised during the early life period of the laying hen chicks, as they lack the physical and motor skills to access the elevated surfaces of the aviary [24–26]. Producers also generally confine the chicks to the lowest aviary tier for the first two to four weeks of age (WoA), as they want the chicks to stay near food, water, and heat sources, thereby restricting the access to complex environments in their early life [27]. To facilitate access to elevated surfaces in the early life of domestic chicks, studies have investigated the provision of ramps, which provide a continuous path connecting the different tiers of the aviary [28,29]. Providing ramps has been shown to facilitate the chicks' access to the elevated surfaces from the first WoA in multi-tiered aviaries without any negative consequences on mortality or welfare indicators [28,29].

Facilitating early access to elevated surfaces using ramps might also have positive consequences on the development of spatial cognition of laying hens. Provision of ramps in aviaries might influence the spatial cognition of the birds mainly in two ways. First, ramps provide earlier exposure to vertical complexity. Laying hen chicks reared with ramps have been shown to use elevated surfaces earlier compared to birds reared without ramps [28,29]. Early exposure to vertical complexity aided by ramps might provide earlier opportunities for spatial learning, which in turn can influence cognition and neural development. Second, ramps promote early life voluntary exercise in domestic chicks. Chicks perform more vertical movements between the tiers of aviaries when provided with ramps, with most of these movements happening

through ramps [28]. Ramp use is not limited to tier change; chicks also walk, run, jump to and from, and perform wing-assisted inclined running on ramps from at least three days of age (DoA) [30]. The increase in early life voluntary exercise has been shown to improve adult hippocampal neurogenesis and spatial memory in rodents [31–33], humans [34,35] and fish [36–38]. Although not directly investigated in hens, it is reasonable to assume that early life voluntary exercise facilitated by ramps would positively influence the development of spatial cognition of laying hens.

Individual differences in laying hens and their importance in cognition have received a recent increase in interest [39,40]. Individual hens differ in the vertical movements and use of resources including the elevated surfaces within the aviary [41,42]. Consistent differences in behaviours such as exploration [43,44] and activity [41], have been established in domestic hens, which has been shown to influence the movement behaviour in various species [45,46]. It is possible that these might drive differences in ramp use among individuals, potentially leading to individual developmental trajectories for spatial cognition. For example, individual hens differ in the time spent on the free range which has been shown to positively correlate with cell proliferation in the rostral hippocampal sub-region [47]. Hence, it is important to account for individual differences in ramp use behaviour to better understand the impact of early life ramp use on the development of spatial cognition in laying hens.

The holeboard test has been used to assess the spatial discrimination learning abilities in various species [48], including domestic hens [23,49]. For the test, a subset of all the potential sites is food-rewarded and a bird can visit any of these sites in whatever order it chooses. The holeboard test provides measures for both short and long-term spatial memories. The reference memory is an indicator of long-term memory and holds trial-independent information such as the locations of food rewards and how to access these food rewards. The working memory and general working memory are measures of short-term memory and hold information that is relevant only for a specific trial such as the sites a bird has already visited in that trial.

To the best of our knowledge, there are no studies that have looked at the relationship between individual differences in ramp use and spatial cognition of laying hens. In the current study, we planned a first exploration of the potential relationship between the early life ramp use and spatial memories of laying hen pullets using a holeboard test.

## Methods

### Ethical approval

The experiment met the federal and cantonal regulations for the ethical treatment of animals involved in research and was approved by the Veterinary Office of the Canton of Bern, Switzerland (BE 106/19).

### Animals and housing

Sixteen custom-built, identical pens (2.00 m × 2.00 m × 2.50 m (L × W × H)) situated at the Aviforum research facility, Zollikofen, Switzerland were used to house a total of 351 Lohmann Selected Leghorn (LSL) chicks from one day of age (DoA) until 17 weeks of age (WoA). All pens housed 22 birds, except for one pen which only had 21 chicks, due to a mistake in chick delivery Each pen was fitted with two vertically stacked tiers at heights of 0.25 m and 1.20 m from the floor, and two round metal perches (2.00 m × 0.34 cm (L × diameter)) at 0.28 m and 0.55 m above the second tier. The tiers in each pen were connected by two ramps (1.30 m × 0.24 m (L × W), at an angle of 35˚) made of metal grids (0.65 m × 0.25 m (L × W)). Both tiers were made of plastic grids with the first tier (2.00 m × 1.15 m (L × W)) being wider than the second (2.00 m × 0.60 m (L × W)). The pen floors were covered with wood shavings (2.00

m × 0.75 m (L × W)). The pens were visually isolated from each other using opaque metal sheets that covered up to 1.50 m of the pen walls from the floor. The rest was covered with plastic sheets. The schematics of the pens and more description can be found in Johny et al. (2023). Olfactory and auditory contact was still possible between pens. Birds were populated on the first tier and confined there using wire mesh until four DoA. This was done to ensure that the birds learn the location of the drinkers and feeder. Feed was provided *ad libitum* on a feeder plate located on the first tier until two WoA, after which the feed was provided from a dispenser placed in the litter area. An additional feeder plate (40 cm diameter, 5 cm height) on the second tier was provided until five WoA. Water was provided *ad libitum* using nipple drinkers on the first tier. The birds were fed with starter feed (Egli Mühlen AG, Nebikon, Switzerland) from one until nine WoA and pullet feed (Egli Mühlen AG, Nebikon, Switzerland) from nine until 17 WoA. A light bulb (Silox basic, 120 – 3000K) attached to each pen ceiling ensured uniform illumination of 40 lux throughout the observation period. There was no provision of natural light. The light schedule followed standard rearing management for LSL (27) pullets with 24 hours of light for the first two days of life, which was gradually reduced to nine hours in the fifth WoA and stayed the same until the conclusion of the project. The dimming phase of the light lasted for five minutes at dawn and 30 minutes at dusk. Birds were vaccinated according to the standard rearing management protocol for LSL birds (Infectious bronchitis-primer, Marek's disease, Gumboro at one DoA at the hatchery, Paracox at five DoA, Infectious bronchitis 4/91 at 16 DoA, Infectious bronchitis Ma5 at 65 DoA, Avian encephalomyelitis at 93 DoA, Poulvac E.coli at 103 DoA and inactivated Infectious bronchitis vaccine at 114 DoA).

During population at one DoA, eight birds per pen were arbitrarily chosen as focal individuals. Each focal individual was marked with animal marking spray (RAIDEX, Hauptner, Switzerland) and tagged with numbered leg rings that were 5 mm in diameter (Flexi-Ringe, Fieger AG, Switzerland). We used green, black, and brown colours to mark the birds. Green and brown colours were used to mark two focal birds per colour per pen by applying a dot either on the back or neck region. Black colour was most recognizable on the chicks especially at younger ages and was used to mark four focal birds per pen. Four patterns were used with black colour–one dot on neck or back region, one dot on both neck and back, and a continuous line that ran from the neck to back. The colour marks were reapplied at 3, 8, 15, 29 and 43 DoA. The leg rings were changed to 8, 12 and 14 mm in diameter at 8, 29 and 43 DoA, respectively. No aggression towards focal birds was observed and no difference in body mass was found between focal and non-focal birds at 14 WoA.

Birds used in the current experiment were also part of another experiment that investigated artificial cues to encourage early life ramp use in laying hen chicks (30). This experiment had four treatment groups, that investigated the use of artificial cues in which three were exposed to artificial cues while one served as a control group devoid of any cues. Because this other experiment's outcome indicated an impact of the treatment on ramp use, we mitigated the influence of treatment group-related variance by incorporating it as a control variable in the mixed-models used for statistical analyses.

## Data collection

**Ramp use.** Each pen was fitted with a video camera (Samsung SCO-2080R, IR, Samsung Techwin CO., Korea) connected to customized recording software (Multieye Hybrid Recorder Version 2.3.1.8, Artec Technologies AG, Diepholz, Germany) that recorded the videos at regular intervals until nine WoA. We assessed behaviours of focal birds (n = 8/pen) by counting inter-tier transitions and active behaviours on ramps, as described in Johny et al. [30] Inter-tier transitions were when birds moved between tiers using ramps. Active behaviours on

ramps included walking, running, wing-assisted incline running (WAIR), or jumping, not resulting in a transition. We didn't differentiate between upward and downward transitions, and counted active behaviours as separate events if they were at least five seconds apart.

Both behaviours were counted from recorded videos at 6, 10, 12, 20, 27, 41, and 55 DoA for eight observation bouts of three minutes dispersed evenly across the light period (total observation time per individual per day = 24 minutes).

**Holeboard test.** *General principles.* We used the holeboard test to measure the reference, working and general working memory of the birds in three different phases following initial habituations to a food reward and the test arena. The first phase was cued and served to test the birds' ability to learn the location of three rewarded cups out of a total of eight cups with colour cues added to the rewarded cups. The second phase was uncued and involved learning the same rewarded locations as in the cued phase, but without the provision of colour cues. In the third, reversal phase, the locations of the rewarded cups were changed, and the birds had to replace the information of location of the rewarded cups they had acquired during the cued and uncued phase with a new one. There were no colour cues associated with rewarded cups in the reversal phase. The birds received 10, 20, and 10 trials each to learn the spatial configuration of the food reward for the cued, uncued, and reversal phases, respectively.

*Arena and procedure.* We tested three focal birds per pen (n = 48) from 56–95 DoA in the holeboard test. The arena design and testing methods for the holeboard test were adapted from Dumontier et al. [50]. Two arenas were used to test two birds simultaneously which were situated in a separate experimental room adjacent to the barn. The arenas were enclosed by 2 m high walls made of tarpaulin on all sides and had a door on the side (Fig 1A). Hence, birds were visually but not auditorily isolated during the testing procedure. On the floor of each arena, eight circles distributed equally spaced in a 2 × 4 matrix were drawn with black permanent markers (Fig 1B). In the centre of each circle, a small cup attached to a plywood platform was placed and was used to provide grape pieces as the food reward during the test. Each arena had a start box that was connected to a pulley system allowing the observer to lift both start boxes simultaneously. Both arenas were equipped with a camera (Samsung SCO-2080R, IR,

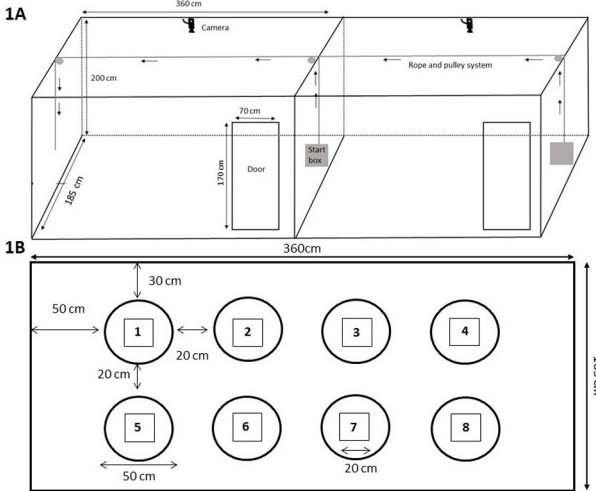

**Fig 1. Schematic representation of the holeboard arena.** (1A) Two birds were tested simultaneously. The black arrows represent the direction of movement of rope and pulley system used to lift the start boxes. (1B, adapted from Dumontier et al [50]. The dimensions of the arena and the general spatial configuration of the cups as viewed from above. The cups were numbered from 1 to 8. The rectangles represent the platforms of the cups.

Samsung Techwin CO., Korea) connected to computer screens located outside the arena but within the experimental room allowing simultaneous viewing, recording, and coding of birds' behaviour by two researchers while birds were tested in parallel using both arenas. The researchers were trained on holeboard test videos obtained from Dumontier et al. [50] to establish inter-observer reliability before the experiment began. For the whole procedure, home pens were randomly assigned to an arena and observing researcher. Once assigned, the birds were always habituated and tested in the same arena and handled and scored by the same researcher (i.e., 8 pens or 24 birds assigned per arena and researcher). The number of grape pieces eaten and each visit of the hen to a cup were scored. If at least half the body of the hen was inside the circle and the head was oriented towards the cup, it was considered a visit.

*Habituation*. The focal birds were habituated to grape pieces in their home pens using the same cups as in the test. The habituation was performed over two consecutive days, with three sessions of 20 minutes per day. After habituation to the food reward, six focal birds per pen were arbitrarily chosen and habituated to the test arena in groups of three over three sessions. The first session lasted for 10 minutes and the other two was five minutes each. Each cup contained three grape pieces during the group habituation. Following the group habituation, the birds were individually habituated to the arena. Each cup contained one grape during the individual habituation. After three individual habituation sessions of five minutes each, three birds per pen that ate the most grape pieces over the three habituation sessions were selected for the test. The selected birds were further habituated to the arena until they ate from all the cups. The holeboard test was conducted in three separate phases as follows:

*Cued phase (trials 1–10)*: Three cups out of the eight cups were baited with a piece of grape. To control for the odour cues, a piece of grape was placed in a hidden compartment below the bottom of each unbaited cup. To provide cues associated with the food reward, the plywood platforms were red in colour for the baited cups, while the unbaited cups had no colour cues associated with them. Three configurations of the baited cups (i.e., the spatial position of the baited cups in the arena) were randomly chosen and assigned to the three focal birds selected for the tests from each pen (i.e., one configuration per bird). The birds were tested in a random order on each test day. At the start of each test, the birds were placed under the start boxes in both arenas by the researchers and doors of the arenas were closed. The test began when the start boxes were lifted by one of the researchers and ended when both birds had consumed the food reward from all three cups, or a maximum of five minutes had elapsed. The birds were returned to the home pen after each trial and were given 10 trials over five days to learn the configuration of the rewarded cups.

*Uncued phase (trial 11–30)*: The same reward configuration and test procedure were used for both the cued and uncued phases, but the baited cups no longer had any visual cues in the latter as the plywood bases of all cups looked the same. The birds were given 20 trials over 10 days to learn the reward configuration without visual cues.

*Reversal phase (trial 31–40)*: To get to the reversal phase, birds had to reach a success criterion calculated as the ratio of number of visits to baited cups to total number of visits to all cups. The ratio indicates the ability of the bird to distinguish between rewarded and unrewarded cups and can be used as a measure of learning performance [48]. Birds that had reached a score of 0.6 or more in three out of four consecutive trials in the uncued phase were subjected to the reversal phase. The reversal phase lasted for 10 trials over five days and the birds had to find the food reward in a new configuration of uncued, baited cups thus measuring the ability of the birds to unlearn the old and learn the new reward configuration. Three new reward configurations were randomly chosen and assigned to the birds that met the success criterion from each pen (i.e., one configuration per bird). Some pens had less than three birds as not all birds met the success criterion.

*Data processing*. For each trial, we performed the following calculations:

1. Reference memory errors: the sum of all visits and revisits to unrewarded cups. The reference memory errors indicate the ability of the birds to discriminate between rewarded and unrewarded cups.

2. Working memory errors: the revisits to rewarded cups (calculated as the difference between the number of rewarded visits and total number of visits to rewarded cups). It reflects the ability of the chickens to avoid re-visits to rewarded cups during a trial.

3. General working memory errors: total number of revisits to all the cups (calculated as the difference between the number of unique visits to rewarded and unrewarded cups and total number of visits to all cups). This indicates the ability of the chickens to avoid re-visits to already visited cups during a trial.
For all memory errors, greater values indicate a poorer performance, with zero being the best possible performance.

4. Trial duration: total time taken in seconds to consume the food rewards from all three cups. Duration was measured starting from the lift of the start box with a maximum of five minutes in case the birds failed to consume the rewards from all three rewarded cups.

5. Errors during phase change: to investigate the response of the birds to the change in phases (i.e., cued to uncued, uncued to reversal), we compared the number of memory errors of each bird from three trials before and after a phase change for all the three memory indices.

## Statistical analysis

All analyses were performed with R (version 4.1.1, R Core Team, 2021) and R studio (RStudio Team, 2021) as the graphical interface. We used 'lme4' [51] package to fit linear mixed models (LMM) and package 'glmmTMB' [52] to fit generalized linear mixed effects models (GLMM). The model assumptions for homogeneity of variance and normal distribution of errors were checked using the 'Dharma' [53] package. When the model assumptions were not met, the data were transformed. Continuous explanatory variables were centred to zero and categorical variables were sum-contrast coded to set the reference level of contrast calculation as the mean of all groups within a variable. This was done to obtain interpretable main effects, even in the presence of interactions. The p-values of each explanatory variable were calculated by reducing each model by the particular explanatory variable and comparing it to the full model using parametric bootstrap tests (package 'pbkrtest' [54]). The model estimates and confidence intervals were calculated using 'broom:mixed' package [55] and the package 'emmeans' was used to calculate estimated marginal means from the full models [56]. We used the framework provided byBerner and Amrhein [57], which uses a combination of effect sizes, confidence intervals and p-values to interpret results. To reduce the numbers of tests, post-hoc comparisons were done qualitatively using numeric and visualizations of the estimated marginal means while also considering the raw data. The 'tidyverse' package [58] was used for data cleaning and the package 'ggplot2' [59] for data visualization.

**Frequency of ramp use.**   We summed the number of transitions and active behaviours on ramps for individuals to obtain total ramp use per bird. We analysed the general ramp use behaviour of all focal birds as well as only the focal birds that underwent the holeboard test descriptively.

**The association between holeboard measures and ramp use.**   For the cued and uncued phase, data from all focal birds that underwent the holeboard task (n = 48) and for reversal

phase the data from focal birds that met the success criteria (n = 37) were included in the analysis. The reference, working and general working memory errors, and trial duration for each bird and trial were used as response variables and were analysed separately for each phase. To relate ramp use frequency to holeboard parameters, the mean ramp use frequency (MRF) was calculated by taking mean ramp use over all observation days per focal bird and used as a continuous explanatory variable. Each model contained MRF per individual and trial number (cued: 1–10, uncued: 11–30, reversal: 31–40) as continuous variables as well as their interaction as explanatory variables. Configuration was added as a control variable in each model, as preliminary visualizations revealed differences in holeboard measures between different configurations. Trial number nested in bird ID nested in pen was used as random factor in all models. The models were fit with Poisson distribution for cued and reversal phase for reference memory errors, cued phase for working memory errors, and reversal phase for general working memory errors. Uncued phase of reference memory errors, cued and reversal phase of working memory errors, and cued and uncued phase of general working memory errors were fit with a negative binomial distribution. All phases of trial duration were fit with a Poisson distribution.

The effect of phase change on reference, working, and general working memory errors were analysed separately for both phase changes. The trials and birds for the phase change analysis were chosen as per point five under the 'data processing' section. All models included an interaction of MRF and respective phases (for ex., cued and uncued for cued to uncued phase change analysis) as categorical explanatory variables. Reward configuration was added as a control variable and trial number nested in bird ID nested in pen was included as a random factor in all models.

The data and code for all analysis can be found at https://osf.io/9j2g8.

## Results

### Ramp use

The mean individual ramp use over 56 observations (eight bouts per seven observation days) was 0.39 uses of ramps per observation bout (i.e., per three minutes) for all focal birds (n = 128). The mean individual ramp use per bout ranged from 0.09 to 1.00 with a standard deviation of 0.16. Of the 7,186 observations, 79% of the observations were zeroes (5,698).

The mean individual ramp use of focal birds (n = 48) that underwent the holeboard test was 0.38 uses of ramps (range 0.09–0.67) per observation bout with a standard deviation of 0.14. Eighty percent (2,153 observations) of the total 2,688 observations were zeros.

### Holeboard test

**Reference memory errors.**   Birds with higher MRF demonstrated a decrease in errors compared to birds with lower MRF during the cued phase, although statistical evidence was limited (model estimates (Est) ± CI = 0.94 [0.88, 0.99], p = 0.08, Fig 2). For example, the bird with the lowest MRF made slightly more errors [Estimated marginal means (EMM) ± CI = 3.36 [2.52, 4.48]) than the ones with median (2.95 [2.61, 3.35]) and highest (2.55 [1.91, 3.41]) MRF. The number of errors also decreased with trial number (Est ± CI = 0.75 [0.76, 0.85], p = 0.001) in the cued phase. The interaction of MRF and trial number did not influence the number of errors made in the cued phase (Est ± CI = 0.97 [0.32, 0.92], p = 0.31). In the uncued phase, the number of errors reduced with trial number (Est ± CI = 0.93 [0.89, 1.07], p = 0.001), but there were no influences of MRF (Est ± CI = 1.01 [0.96, 1.07], p = 0.74) or the interaction between MRF and trial number (Est ± CI = 1.00 [0.97, 1.04], p = 0.98). In the reversal phase, the birds with higher MRF made more errors than the birds with lower MRF

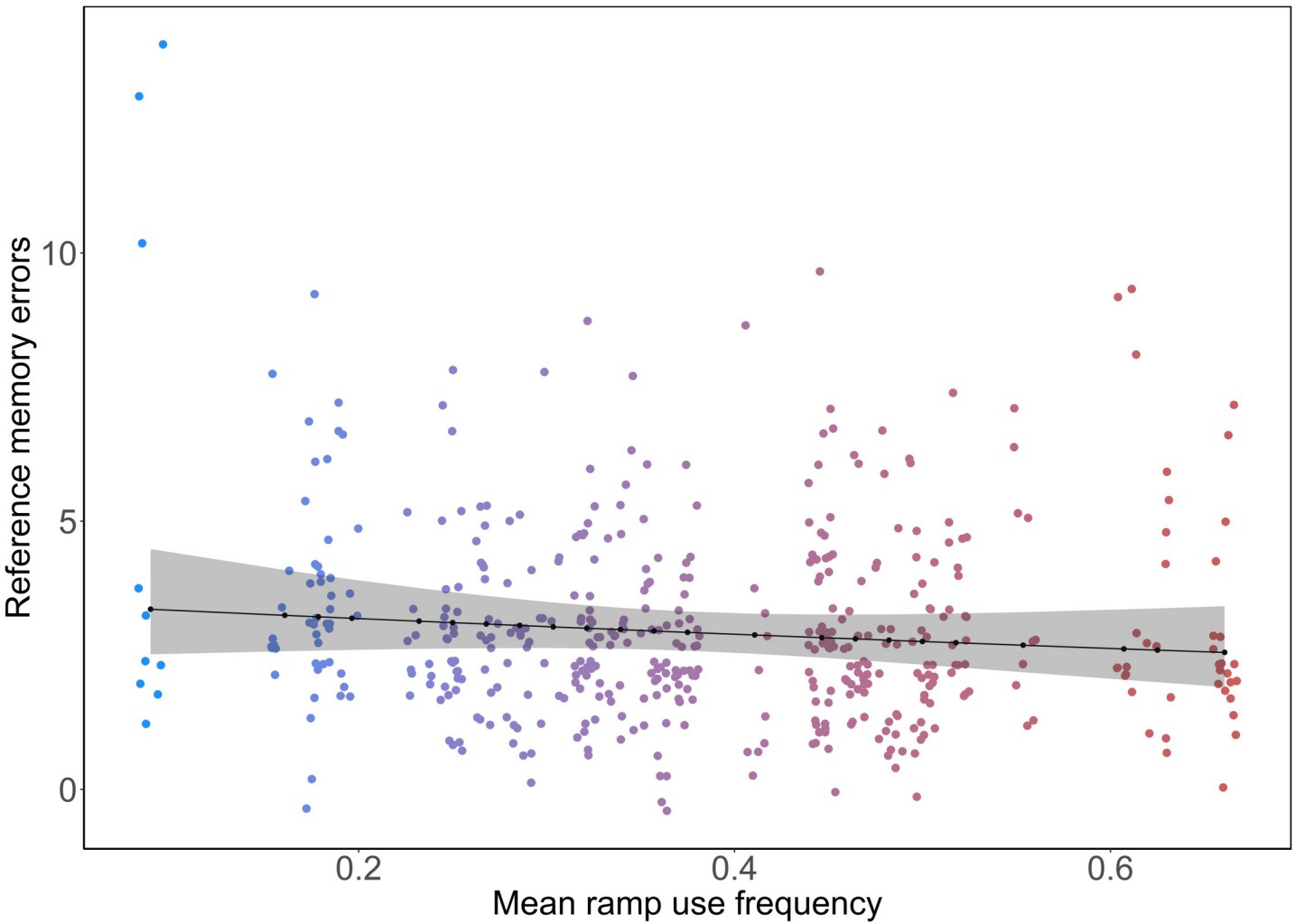

**Fig 2. The relationship between reference memory errors (sum of all visits and revisits to unrewarded cups) and mean individual ramp use frequency in the cued phase (model estimates ± CI = 0.94 [0.88, 0.99], p = 0.08).** The black points connected by the black line represent the estimated marginal means and the shaded ribbon represents 95% confidence intervals. The coloured points represent reference memory errors (all visits–visits to rewarded cups) of focal birds' colour-graded according to mean ramp use frequency.

(Est ± CI = 1.10 [1.01, 1.20], p = 0.05, Fig 3). The bird with the lowest MRF made slightly fewer errors (EMM ± CI = 2.72 [1.82, 4.07]) than the ones with median (3.30 [2.80, 3.90] and high (4.01 [2.78, 5.77]) MRF. With limited statistical support, the number of errors slightly decreased with trial number (Est ± CI = 0.95 [0.9, 1.01], p = 0.08) in the reversal phase. The interaction of MRF and trial number did not have any effect on the number of errors made in the reversal phases (Est ± CI = 0.99 [0.94, 1.04], p = 0.82).

**Working memory.**  In general, all birds made very few working memory errors throughout all phases (Mean ± SD: cued– 0.33 ± 0.69, uncued– 0.23 ± 0.64, reversal– 0.34 ± 0.74). With weak statistical support, the birds with higher MRF made marginally fewer errors than birds with a lower MRF (Est ± CI = 0.77 [0.59, 1.00], p = 0.06, Fig 4) in the cued phase. The bird with the lowest MRF (EMM ± CI = 0.47 [0.15, 1.45]) made slightly more errors than the ones with median (EMM ± CI = 0.27 [0.15, 0.47]) and the one with the highest MRF (EMM ± CI = 0.15 [0.05, 0.52]). There was a slight reduction in the number of errors as the trial number increased, although the statistical support for this find was weak (Est ± CI = 0.84

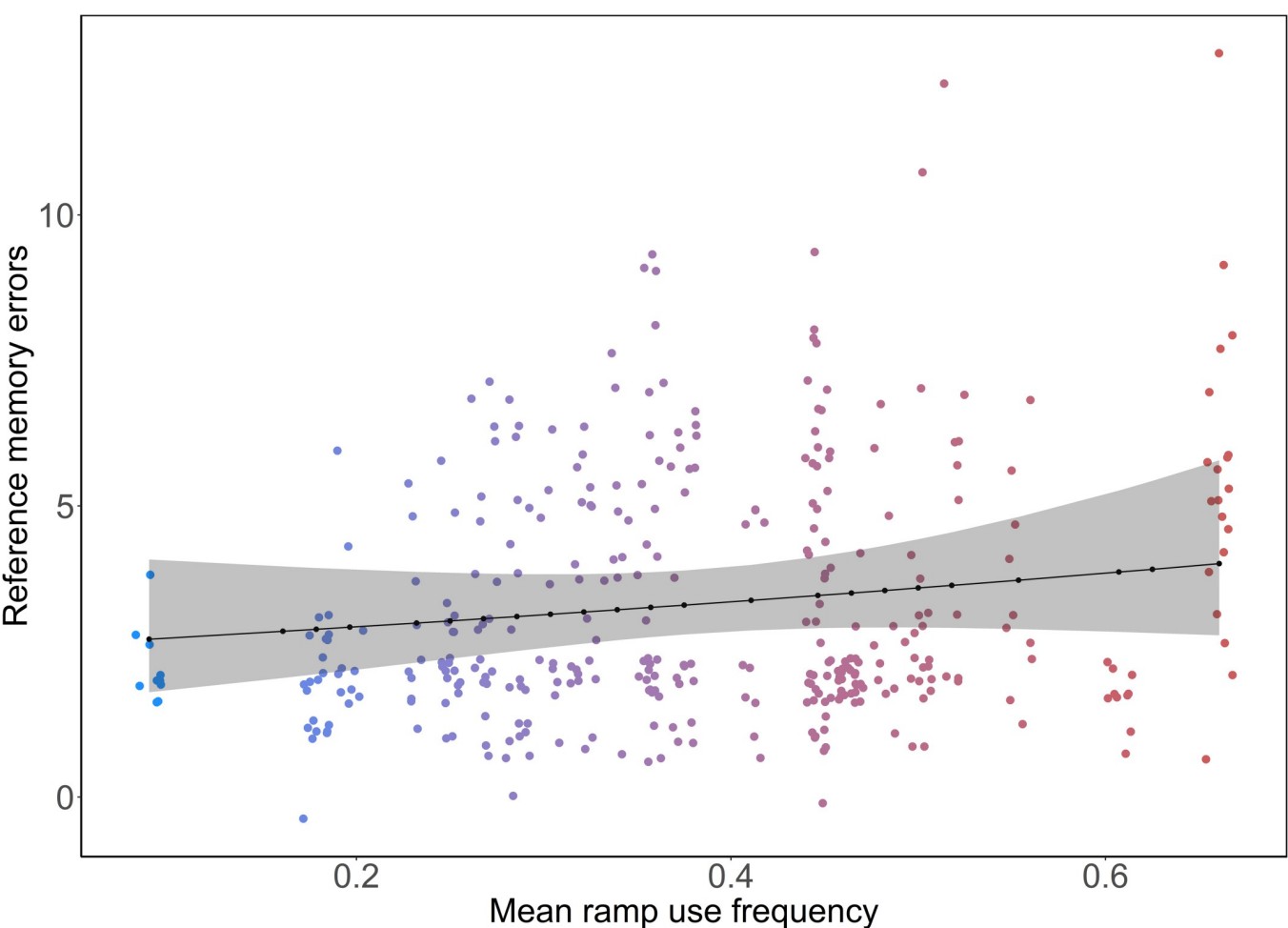

**Fig 3. The relationship between reference memory errors (sum of all visits and revisits to unrewarded cups) and mean individual ramp use frequency in the reversal phase (model estimates ± CI = 1.10 [1.01, 1.20], p = 0.05).** The black points connected by the black line represent the estimated marginal means and the shaded ribbon represents 95% confidence intervals. The coloured points represent reference memory errors (all visits–visits to rewarded cups) of focal birds, colour-graded according to mean ramp use frequency.

[0.69, 1.03], p = 0.08). The interaction between trial number and MRF did not a have an effect number of working memory errors in the cued phase (Est ± CI = 0.94 [0.78, 1.15], p = 0.80). In the uncued phase, the number of errors decreased with trial number (Est ± CI = 0.87 [0.76, 0.99], p = 0.05). No influence of MRF (Est ± CI = 1.07 [0.86, 1.39], p = 0.12) or the interaction between MRF and trial number (Est ± CI = 0.95 [0.84, 1.08], p = 0.45) was found on the number of errors made in the uncued phase. In the reversal phase, the number of errors made by the birds was not influenced by MRF (Est ± CI = 1.03 [0.78, 1.36], p = 0.87), trial number (Est ± CI = 0.83 [0.67, 1.06], p = 0.14) or their interaction (Est ± CI = 0.84 [0.66, 1.05], p = 0.14).

**General working memory.** Birds made very few general working memory errors in all three phases (Mean ± SD: cued– 0.73 ± 1.56, uncued– 0.62 ± 1.55, reversal– 0.86 ± 1.61). The number of errors made by birds decreased with trial number with varying statistical support for each phase: cued (Est ± CI = 0.69 [0.56, 0.84], p = 0.003), uncued (Est ± CI = 0.85 [0.72, 1.04], p = 0.08) and reversal (Est ± CI = 0.87 [0.78, 0.97], p = 0.02). No influences of MRF (Est ± CI, cued = 0.80 [0.61, 1.04], p = 0.12, uncued = 1.10 [0.86, 1.39], p = 0.46, reversal = 1.17

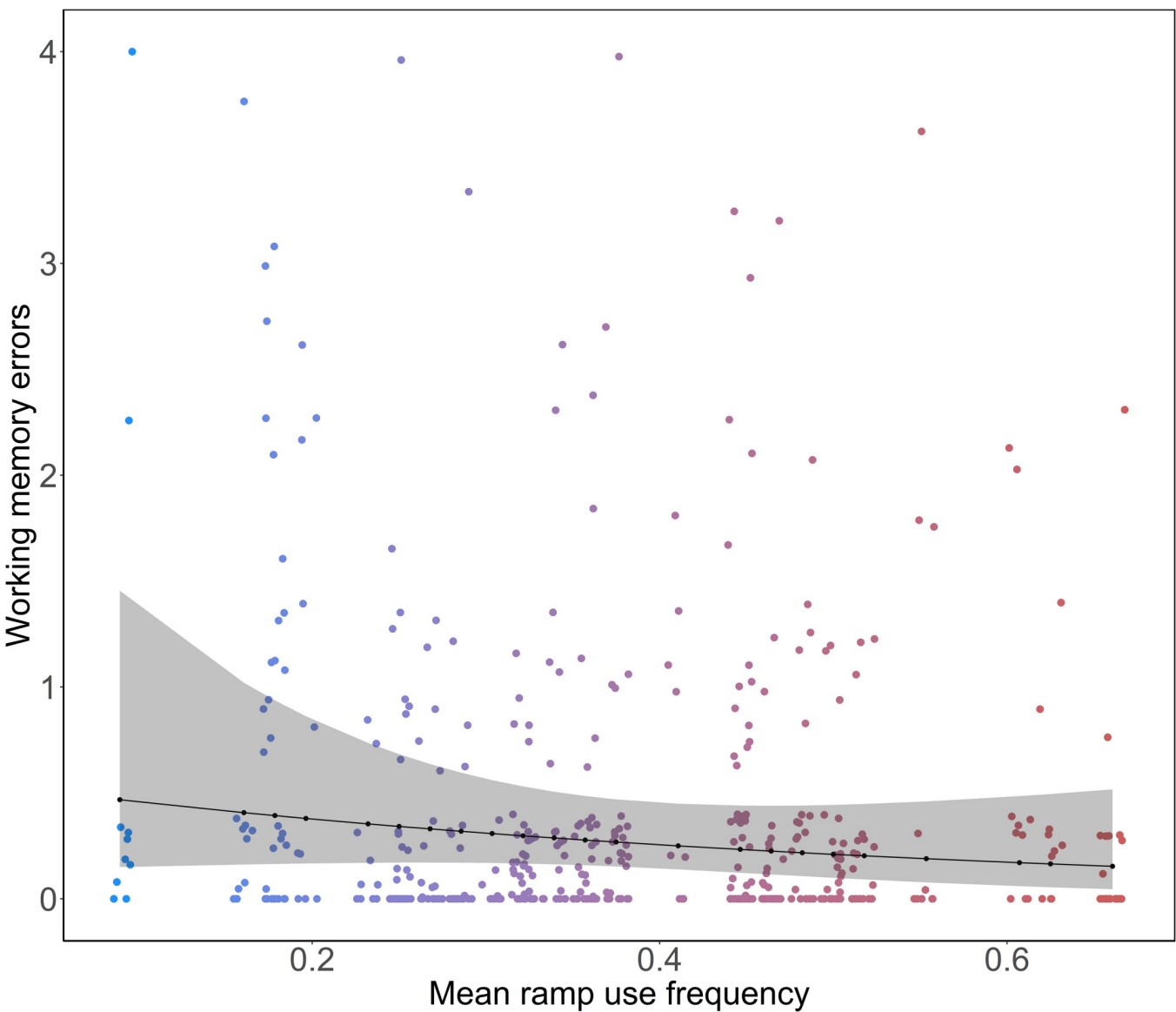

**Fig 4. The relationship between working memory errors (sum of all revisits to rewarded cups) and mean individual ramp use frequency in the cued phase (model estimates ± CI = 0.77 [0.59, 1.00], p = 0.06).** The black points connected by the black line represent the estimated marginal means and the shaded ribbon represents 95% confidence intervals. The coloured points represent working memory errors (all visits to rewarded cups–rewarded visits) of focal birds, colour-graded according to mean ramp use frequency.

[0.90, 1.53], p = 0.30) and the interaction between MRF and trial number (Est ± CI, cued = 0.89 [0.70, 1.10], p = 0.28, uncued = 0.99 [0.83, 1.19], p = 0.93, reversal = 0.94 [0.84, 1.06], p = 0.30) were found for any of the phases.

**Trial duration.**    The trials in which the birds failed to consume from all baited cups were removed from the analysis (9, 4 and 1 out of 480, 960 and 370 observations from cued, uncued and reversal phases, respectively). Trial duration was influenced by an interaction between MRF and trial number (Est ± CI = 0.88 [0.84, 0.93], p = 0.001) in the cued phase, with birds with a high MRF showing a faster decline in trial duration with increasing trial number (Estimated slopes ± CI for selected MRF: lowest = - 0.19 [- 0.24, - 0.13], median = - 0.29[- 0.32, -0.27], highest = - 0.40 [- 0.45, −0.34]). Trial duration decreased with increasing trial number

in the uncued (Est ± CI = 0.99 [0.98, 1.00], p = 0.001) and reversal (Est ± CI = 0.93 [0.91, 0.95], p = 0.001) phases. No effect of MRF (uncued = 1.05 [0.53, 2.02], p = 0.33, reversal = 0.18 [0.02, 2.03], p = 0.20) nor the interaction of MRF and trial number (uncued = 1.00 [0.99, 1.02], p = 0.78, reversal = 1.07 [1.01, 2.03], p = 0.79) were found for trial duration for the uncued and reversal phases.

**Phase change.** The interaction between phase change and MRF) had an effect on the reference memory errors (p = 0.001) for the cued to uncued phase change with birds with a higher MRF making more errors than the birds with a lower MRF as the phase changed from cued to uncued (Fig 5A). For the uncued to reversal phase change, there was no effect of the interaction between phase change and MRF on the reference errors (p = 0.17). The interaction between phase change and ramp use group had no effect on the working memory scores for both phase changes (cued to uncued, p = 0.21, uncued to reversal, p = 0.32). For the general working memory, the interaction between phase change and ramp use group showed an effect for the cued to uncued (p = 0.02), but not for the uncued to reversal phase change (p = 0.16). The birds with the lower ramp use made marginally more errors than the ones with higher ramp use as the phase changed from cued to uncued for the general working memory (Fig 5B).

## Discussion

The aim of the study was to perform a first exploration into the relationship between early life ramp use (1–8 WoA) and short and long-term spatial memory in laying hen pullets. We found that the birds that used ramps more made fewer reference and working memory errors in the cued phase than birds with lower ramp use. However, birds with higher ramp use made more reference memory errors in the reversal phase. The birds that used ramps more often also showed a faster decline in time taken to solve the holeboard test with increasing trial numbers in the cued phase. For the phase change analysis, we found that birds with a higher ramp use made more reference and general working memory errors in the cued to uncued phase change than the birds that used ramps less often. Our results indicate a relationship between early life ramp use and spatial memories in laying hens, which can have several explanations.

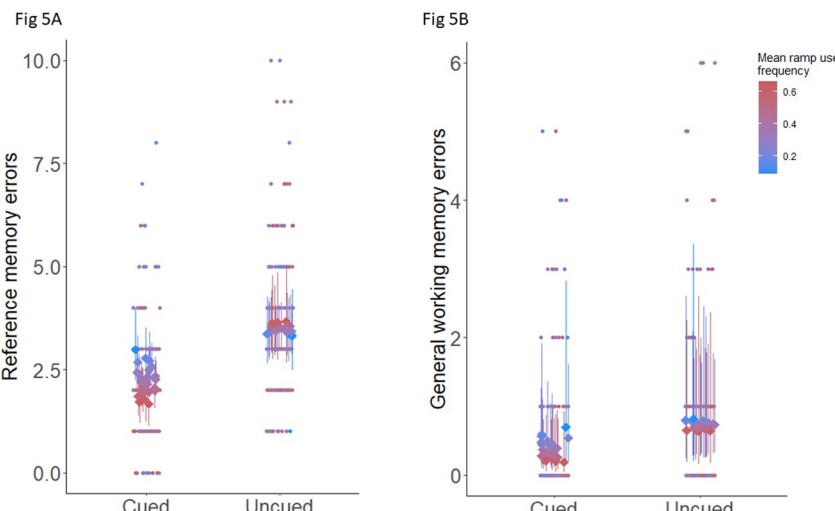

**Fig 5.** The relationship between ramp use frequency and spatial memory errors (A) Reference memory, B) General working memory) as the phase changed from cued to uncued phase. The point ranges represent the estimated marginal means of reference memory errors and the points are the raw data, colour graded according to mean ramp use frequency.

### The relationship between early life ramp use and spatial memories

One possible explanation for the association between holeboard measures and ramp use might be that the individual differences in both ramp use and cognition are driven by consistent inter-individual differences in behaviour. Consistent differences in behaviours such as boldness, exploration and activity have been shown to correlate to movement in animals [46,60]. For example, bolder bank voles [61] and Trinidad killifish [62] move longer distances, fast exploring female great tits disperse to a greater distance [63], and high exploratory laying hens use the outdoor range more than the ones that are less exploratory [64,65]. It is possible that individual differences in ramp use might also be driven by individual differences in activity, exploration, and fearfulness, with more active, explorative, and less fearful birds showing more use of ramps.

The speed-accuracy trade-off, proposed by Sih and Del Giudice [66], is a key framework for understanding consistent inter-individual differences in behaviour and cognition. It suggests that fast behavioural types (with high exploration and activity levels, neophilia, etc.) prioritize speed over accuracy, acquiring simple rules quickly but making more mistakes in challenging situations like reversal learning. Conversely, slow behavioural types (with low exploration and activity levels, neophobia, etc.) prioritize accuracy, and while they are slower at simple learning tasks, they are more sensitive to changes in their environment and make fewer errors during reversal tests. While not studied in domestic hens, our results seem to align with this model. Birds with higher ramp use made fewer spatial memory errors and learned tasks faster in the cued phase, suggesting better associative learning [67]. In contrast, birds with lower ramp use made fewer errors during the reversal phase, indicating higher behavioural flexibility [68]. They also showed fewer memory errors during phase change from cued to uncued, suggesting an ability to update learning strategies with environmental shifts. Additionally, the marginally higher working memory errors of birds with low ramp use in the cued phase may indicate neophobia, as cues were absent during habituation.

Empirical research however indicates that the relationship between consistent inter-individual differences in behaviour and cognition does not always follow the speed-accuracy trade-off framework and that the relationship is dependent on factors such as species, sex, and age [69]. For example, Zidar et al. [44] showed a complex and age-dependent relationship between cognition and behavioural type in red jungle fowl. As red jungle fowls and domestic chicks are closely related, we may need further studies across multiple ages to elucidate the complex relationship between consistent inter-individual differences in behaviour and cognition in laying hens.

Another possible explanation of our results might be that early life ramp use did not have a meaningful influence on the spatial memories of the hens, a position supported by the low effect sizes (or influence) of MRF on spatial memories (Reference memory, cued phase: 0.94, reversal phase: 1.10). Moreover, we also didn't see any differences in the spatial memories in the uncued phase, which would be expected if the birds with different ramp use varied in their spatial cognition as well. As described within the context of the speed-accuracy trade off above, the differences in spatial memories observed in the cued and reversal phase can be explained by a difference in associative learning and behavioural flexibility, respectively. In comparison to the cued and reversal phases, the performance in the uncued phase is directly related to spatial memories and has been shown to be controlled by the hippocampus [48], the region that plays a central role in processing spatial information [9]. Hence the lack of differences in spatial memories in the uncued phase, combined with low effect sizes of reference memories in the cued and reversal phase suggests that early life ramp use might only have a minor impact on the spatial cognition of the pullets.

## Evaluation of holeboard task

Unlike in rodents and pigs, the holeboard task is not widely used to measure the spatial cognition of hens and hence require a through consideration of the variations and limitations of the methods. Our holeboard test setup was similar to Dumontier et al. [50], except for the lack of two start box positions in the arena and the order of the phases as it was unued, cued, over-training and reversal phase for Dumontier et al. [50]. Also, instead of including an overtraining phase as used by Dumontier et al. [50], we used a success criteria to make sure the birds had learnt the spatial location of the rewards before continuing with the reversal phase. We calculated memory errors rather than memory ratios like Dumontier et al. [50], because of the low variation in the working memory and general working memory ratios in our study. Our birds showed relatively high working (mean ± SD = 0.94 ± 0.13) and general working memory scores (0.93 ± 13) which were similar to Dumontier et al. [50] (working memory = 0.8–0.9, general working memory = 0.7–0.9), but slightly better than in more complex holeboard tests with a $3 \times 3$ matrix of cups (working memory = 0.7–0.8 [49] and 0.6–0.9 in [23]). The reference memory in our study was 0.55 ± 0.13, which again was similar to other studies (0.4–0.7; (23,44,45)). The similarity in scores with other studies, coupled with the decrease in reference memory errors throughout all phases suggest that the birds did learn the task.

As noted by Dumontier et al. [50] and Tahamtani et al. [23], one general criticism of the holeboard test as a method might be the low cost to check the unrewarded cups. There was only a marginal decrease in reference memory errors in the uncued phase from trial 11 to 30. The reference memory errors also showed a large variation across consecutive trials, which might be due to birds checking the all the cups in certain trials, as it incurs only a negligible energy expenditure. If indeed the birds were checking all the cups in certain trials, the lack of large differences in references memory errors observed in our study might not be indicative of lack differences in spatial memory. Future studies employing the holeboard test might benefit from adding additional costs, such as bigger arenas with higher proportion of unrewarded holes or birds required to manipulate the hole, to obtain rewards. For instance, holeboard tests for pigs employ a ball to cover the food bowl, which the pigs have to manipulate using their snout to gain access to rewards underneath [70].

Another limitation specific to our study is the use of one start location throughout the test, which can result in egocentric approaches such as fixed food search patterns [48]. In addition, the rectangular arena with a 2×4 matrix distribution of rewards, can lead to fixed simple food search patterns such as visiting the closest cup straight ahead.

## Outlook

The current study contributes to the existing body of research exploring the relationship between early-life habitat complexity and spatial cognition in laying hens. Many studies have covered multiple levels of complexity (cage vs aviary [23,50], floor vs three-dimensional rearing [15,71]) and ages for a set of spatial cognitive indices (spatial long and short-term memories [23,50], depth perception [71], neuro-motor coordination [15]). Nonetheless, the long and short-term impacts of early life habitat complexity on the spatial cognition of laying hens remains ambiguous. Future studies should account for effect of age, type of test used and include additional measures of brain development such as neural cell counts to better understand the impact early life habitat complexity has on the development of spatial cognition in laying hens. Furthermore, it is important to investigate the specific level of habitat complexity required to induce developmental changes in the spatial cognitive abilities of the birds. This aspect deserves further investigation to determine the optimal conditions for promoting spatial cognition in laying hens.

## Acknowledgments

We thank Abdulsatar Abdel Rahman for helping with the holeboard task. Many thanks to Thomas Heinzel and Jan Büchler, for building the pens and arena, Masha Marincek and Josie Siegel for their help with the video analysis. We thank Lucille Dumontier for the discussions on the statistical analysis. We thank the Ana Rentsch for her valuable feedback that improved the quality of the manuscript.

## Author Contributions

**Conceptualization:** Alex Johny, Andrew M. Janczak, Michael J. Toscano, Ariane Stratmann.

**Formal analysis:** Alex Johny, Ariane Stratmann.

**Funding acquisition:** Michael J. Toscano.

**Investigation:** Alex Johny.

**Methodology:** Alex Johny, Andrew M. Janczak.

**Project administration:** Alex Johny, Ariane Stratmann.

**Supervision:** Andrew M. Janczak, Janicke Nordgreen, Michael J. Toscano, Ariane Stratmann.

**Visualization:** Alex Johny.

**Writing – original draft:** Alex Johny.

**Writing – review & editing:** Alex Johny, Andrew M. Janczak, Janicke Nordgreen, Michael J. Toscano, Ariane Stratmann.

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
