## [Decision Letter · Decision Letter 0]

2 Oct 2023

PONE-D-23-28746Mind the ramp: Association between early life ramp use and spatial cognition in laying hen pullets.PLOS ONE

Dear Dr. Johny,

Thank you for submitting your manuscript to PLOS ONE. After careful consideration, we feel that it has merit but does not fully meet PLOS ONE’s publication criteria as it currently stands. Therefore, we invite you to submit a revised version of the manuscript that addresses the points raised during the review process.

We look forward to receiving your revised manuscript.

Kind regards,

Lamiaa Mostafa Radwan, Ph.D.

Academic Editor

PLOS ONE

Journal Requirements:

"The work was supported by the European Union’s Horizon 2020 research and innovation programme under the Marie Skłodowska-Curie grant agreement No 812777. The document reflects only the author's view and the European Union’s Horizon 2020 research, and innovation programme is not responsible for any use that may be made of the information it contains."

"AJ: The work was supported by the European Union’s Horizon 2020 research and innovation programme under the Marie Skłodowska-Curie grant agreement No 812777. The funders had no role in study design, data collection and analysis, decision to publish, or preparation of the manuscript"

**Additional Editor Comments:**

Dear Dr. Alex Johny

Thank you for submitting your manuscript to PLOS ONE. After careful consideration, we have decided that your manuscript needs Major Revision.

Kind regards,

Prof. Lamiaa Mostafa Radwan, Ph.D.

Academic Editor

PLOS ONE

Editor Comments:

1- Discussion of the results requires more depth in interpreting the results

2- The statistical model needs clarification

3- The entire manuscript needs a brief, more in-depth and clearer writing, with recent references cited.

Reviewers' comments:

Reviewer's Responses to Questions

**Comments to the Author**

1. Is the manuscript technically sound, and do the data support the conclusions?

Reviewer #1: Partly

Reviewer #2: Yes

2. Has the statistical analysis been performed appropriately and rigorously? 

Reviewer #1: I Don't Know

Reviewer #2: I Don't Know

3. Have the authors made all data underlying the findings in their manuscript fully available?

Reviewer #1: Yes

Reviewer #2: Yes

4. Is the manuscript presented in an intelligible fashion and written in standard English?

Reviewer #1: Yes

Reviewer #2: Yes

5. Review Comments to the Author

Reviewer #1: This study tested the association between spatial memory and learning (holeboard test) and voluntary ramp use in laying hen pullets. They found no within-bird consistency for ramp use behaviour. They did find relationships between ramp use frequency and the performance in the holeboard test, some of them supporting their predictions and some to the contrary. The experimental design is sound, though the statistical analyses are perhaps unnecessarily complicated. If no statistician was involved, I would recommend that the authors consult a statistician. If there was a statistician involved, then the section on statistics needs a bit of clarification. The manuscript is well written though it is lengthy and unnecessarily wordy and repetitious at times. My biggest concern is the discussion. There appears to be a lack of references and the structure can be improved by reducing redundant elaborations. Once cleaned up, this manuscript can be a valuable contribution to the laying hen literature.

Major Comments

Stay consistent with the numbering (spelling out numbers under 10), please check the whole manuscript for inconsistencies.

Abstract

You need to introduce terminology somewhere: chicks vs hens vs pullets. This journal is not poultry-specific.

L27: what analyses? Give a model or analysis type to help understand the results.

L30-33: These results are a bit confusing. First, they are trends/tendencies and not significant results, so you should state that. Second, it is unclear what the unit of those estimates are.

L37: Please state why this would be relevant/ interesting.

Introduction

L63: I would suggest exchanging the Campbell reference with Widowski and Torrey’s book chapter on rearing young birds for adaptability: https://linkinghub.elsevier.com/retrieve/pii/B9780081009154000038. This statement refers to a shift in research interest based on the shift in management. Both the Ferreira and Widowski and Torrey references are revisions of the literature and the importance of research in this area. The Campbell paper is an experiment that can be mentioned later when discussing specific findings.

L108-116: This has to go before the research question/objectives (L106-107). Then, add lines 106-107 to the beginning of the next paragraph that contains your predictions.

Methods

L130-131: It’s a bit confusing to point to your other paper for the pen schematics. I immediately looked at your Figure 1 and expected to see a pen. I would suggest you provide schematics or a photo here. Alternatively, add the reference after your description and say that more information and schematics can be found in Johny et al. (28). But I would prefer a photo or schematics in this paper.

L157: How did you choose your random sample? How did you make sure there was no bias (e.g., first catches could be bold, last in the box could be most fearful).

L169-173: Unclear what you mean here. What treatments? How were they managed? Why is this important?

L292-295: This section needs clarification. I do not understand it. Do you mean the slopes in your statistical model? I also don’t understand why you group the birds into high and low ramp users. I though your sample was random? Did you have two clusters? Elaborate on this and on why you chose to create these categories instead of using their actual frequencies.

L306-308: Why did you use bootstrapping to get the main effect p-values? To the best of my knowledge, this method is used to reduce a model and get p-values for reduced factors only. The main effects can be assessed as part of the model. Was a statistician involved in this analysis? If so, I will retract my comment. Otherwise, I would like you to consult a statistician.

L326: If you only include non-zero observations, are you not biasing the analysis? What if a bird uses it occasionally? As I understand it, you would overestimate the repeatability by only including the observations where the bird happened to use the ramps and excluding observations where it didn’t.

Results

I recommend making the sentence about the effects of p<0.01 less vague. If you decide to report tendencies (which you need to clarify in the stats section), report them without embellishments. E.g., ‘errors tended to decrease with trial number’ instead of ‘with very limited statistical support, Y made slightly fewer errors’.

L385-386: I am still confused about where these high and low classifications come from… and now there is a medium group. Was I supposed to know that? Maybe I missed it.

L448: I am not sure if removing these data points is appropriate. What if the group you claim learned faster actually had trials in there that were not completed? I recommend analyzing it with the complete data set or declaring which ramp use groups the omitted data points came from if they were exclusively from low ramp users.

Discussion

There is a general lack of references supporting sweeping statements in this discussion. Please put your results into the context of existing literature and how your study adds to the knowledge base. There is also a lot of repetition that could be cut down (e.g., repeating of your own results in every paragraph).

L497-505: But your birds’ ramp use behaviour was inconsistent over time. How would that translate to personality, which is what you are implying but not spelling out here.

L568-572: Why would this only affect the uncued phase? If HR pullets do better in the cued but not the uncued phase, would this mean there is no difference in spatial learning (location) but only in association learning (colour)?

L580-583: WoA 19 is not the start of peak of lay, and WoA25 is not sexual maturity… it would be the reverse. Are you sure these ages and the terminology have the correct references? I also think that these are two different aspects. Sexual maturity would end the ontogenesis and be accompanied by hormonal changes and, in the case of laying hens, also with a change in housing. The peak of lay is well after that and is not initiated by hormonal or housing changes. I would be hesitant to group those two results and make one statement about behavioural consistency during ontogeny.

L590-593: As ramp use was the behaviour in focus, why choose an observation period that you think is too short? If you had good reason to assume it was appropriate before you started, you should not backtrack now but rather accept your results. If you have reason to believe it was inappropriate based on new information after the fact, please add a section with limitations and explain your initial rationale and what changed.

L593-596: No. You tested a hypothesis and did not reject the null hypothesis. If you have reason to assume a type 2 error, you must provide it. Otherwise, you cannot state your initial prediction after not confirming it. Also, what is ‘short-term’ consistency even? If you introduce a vague term like that, you have to define it and back it up with literature.

Minor Comments

Do not be alarmed by the number of comments, most are very small and should be easily addressed.

Abstract

L19: ‘affect the brain’ is very vague. Be mor specific or don’t mention the brain.

L25: ‘observation period’ what ages are those? Instead of observation period. Give an age range.

L25-26: ‘spatial cognition’ be more specific on what aspect of spatial cognition this test is assessing.

L26: The verb refers to spatial cognition not to the birds. Change ‘were’ to ‘was’.

L29: ‘averaged across all observation days’ is redundant as you already stated that you analyzed the means.

L30: spell out Est and CI when you first use it.

Introduction

L67-68: I don’t mean to increase my own citation but there is newer literature on this which also happened to be mine… I’m not telling you to include it, but you might want to consider it. See https://doi.org/10.1016/j.applanim.2023.105997

L69-70: Same comment as above. Consider including this reference as it is more recent: https://doi.org/10.1038/s41598-023-35956-1

L88-91: This sentence is too long and complex and I am not sure what the content is. Not only do they use ramps to transition between tiers but also to… what?

L96: The ‘inter’ part of inter-individual differences is redundant as there can be no intra-individual difference in cognition as far as I’m aware (unless you do a before/after thing). Adding the ‘inter’ suffix draws attention and seems unnecessarily confusing.

L96-97: This sentence is a bit awkward. I would suggest rewording it. Here are two suggestions, but it’s obviously up to you. ‘There has been a recent peak in the interest in individual cognitive differences and their importance in the study of cognition.’ OR ‘Individual cognitive differences and their importance in the study of cognition have received a recent increase in interest/attention.’

L97-98: Add a comma after surfaces

L99-100: Personally, I would say locomotion instead of movement behaviour, but that’s just a suggestion. Or are you talking about types of locomotion?

L99-101: ‘individual differences’ three times in one sentence. I would change the last one to say ‘can lead to differences in developmental trajectories’. The whole sentence is a bit vague. I would say ‘results in’ instead of ‘can result in’ as the start of the sentence already states that this is a possibility and not a fact.

L103: ‘Hence, it is important…’, but that might be a personal preference.

L104: Again, just say individual differences, not individual-level differences (redundant). Or rephrase to say ‘differences at the individual level’.

L106: The ‘inter’ part of ‘inter-individual variation’ is redundant.

L107: Here too, I find that spatial cognition is too broad a term for a holeboard test. Say what aspect of spatial cognition you were testing.

L117-120: this sentence is too long. Break this into two or even three sentences.

Methods

L129-130: Either say ‘All pens housed 22 birds’ or ‘Pens housed 22 birds each’. Using ‘all’ and ‘each’ is too unnecessary.

L136-137: half of this sentence is redundant. The pen floors were covered in wood shavings (dimensions). OR The pen floors were littered with wood shavings (dimensions).

L138: plastic barriers made of metal sheets? Generally, that sentence is a bit wordy. Simplify

L141: why were they confined for the first 4 days? I know why, but you need to spell it out (briefly!) for non-poultry people.

L142: What do you mean by feeder plate? Like a flat dish? Can you give a description?

L147: It would be nice to have the LUX. I’m assuming it changed with age, so you could add it to the sentence where you give the change in light period length.

L151: Wordy -> five minutes at dawn and 30 minutes at dusk.

L151: Why say that daylight was blocked? It would suffice to say there was none. Or even better, add ‘there was no natural light’ to your lighting sentence.

L167: How do you know there was no aggression? Anecdotal observations? Injuries?

L178-179: The software ‘recorded the behaviour’? Did it code it? Did it record locations? How? Did it record birds, and you recorded the behaviour?

L180-186: There are many repetitions here and it’s wordier than necessary. I suggest rewriting this passage, making it more concise. E.g., “…: the inter-tier transitions (up and down combined) and active behaviours on the ramp (walking, running, wing-assisted incline running, jumping to or from the ramp without changing tier; see ethogram in Johny et al (28)).

L207-208: Hence, birds were visually but not auditorily isolated.

L209-211: Don’t repeat the information that is in the figure. The arena held eight circles containing small cups to provide the reward (grape pieces).

Figure 1: It might help to visualize that 1B represents half of 1A. E.g., you could draw lines from the corners of one pen to the corners of the arena.

L220 ff: This should go into the paragraph ‘Cue phase trial’. Continuing to habituation from here seems like a step back. I suggest you rename this paragraph to ‘Arenas’ and only include the arena description. It would make more sense to go from ‘Arenas’-> ‘Habituation’ -> testing procedure.

L234: Did you ensure that your focal birds had eaten the grapes during the habituation period?

L234: How did you choose the focal birds? Convenience sampling?

L254: Did you use a learning criterion, or did all birds eat all grapes in all trials?

L259: I suggest you move the information on the learning criterion of the uncued phase into the section about the uncued phase. Also, If birds reached the criterion early, did they still do all 20 trials?

L283: what is the unit here?

L288 ff: To delete the very long and complicated sentence on L288-292, you could include the number of trials like so: “… , we compared errors of each bird from the three trials before and three trials after a phase change …” This is only a suggestion.

L302-305: Out of curiosity, what was wrong with your main effects before? I have not seen this to be necessary before, does it not over-specify your models?

L332: what treatments? You cannot expect people to read other papers to understand this one.

Results

Furthermore, many repetitions can be avoided by adding units (count/3min). The reader can be trusted to remember that there were 8 bouts per 7 observation days after you mentioned it once.

L365-368: Repetitions and inconsistencies. You are giving the mean, min, and max count/3min for the individual observations but the mean you call a frequency and the min and max you don’t specify. This could be more concise and clarified. Also, you repeat ‘8 bouts per 7 observations days’ which is unnecessary. Furthermore, you are inconsistent with the numbering. It makes the passage more complicated and harder to understand than need be.

L388: I suggest you delete the word ‘seem’. It either influenced it or not. You can even say you didn’t detect an effect of… but be less vague.

Figures 2-3: I recommend only showing the statistically significant results (Figure 3); the others are sufficiently described in the text (also for Figure 4). If you choose to show all plots, I recommend merging 2 and 3 into one plot with two panels, as the caption is the same.

Figures 2-4: I am confused about the colour changes. The grading is according to the Y-axis and does not add information. It made me think there were different treatments.

L459-460: You have been using HR and LR for a long time at this point; no sense in spelling it out in this bracket. The bracket might be superfluous anyway, especially if you added a section explaining how you defined these groups, as I recommended earlier.

Table 1: Please add indicators for significant differences (e.g., superscript letters).

Discussion

L478: specify the aspect of spatial cognition you assessed.

L479: specify the early life period. If you refer to the first eight weeks, move it up from the end of the sentence.

L481-488: Add which results supported your hypotheses and which didn’t.

L491: Add reference supporting your statements.

L518-530: Most of this is not new information. This whole paragraph seems redundant. Add the main points to the last paragraph where needed and delete this paragraph. There is also a lack of references backing your statements.

L587-590: The cattle example seems out of place. I recommend you remove it.

L597-614: This paragraph seems out of place and irrelevant. It adds no new information and does not support your suggested interpretations. I recommend deleting it as it dilutes your discussion.

L615-620: This would be very interesting sections on limitations. However, I am missing a discussion on the consequences of these limitations. How could these limitations affect your results and interpretation?

L627-629: Add references to all of these statements.

L634-636: That was done here: https://doi.org/10.1016/j.applanim.2023.105997 and to a degree here: https://doi.org/10.1038/s41598-023-35956-1.

Reviewer #2: Consideration of welfare in laying hens is good area of research, specially when battery cages are not much in use in Europe. Overall, this is a well written paper, however, it needs some revisions. Some of these are mentioned in the attached file.

6. PLOS authors have the option to publish the peer review history of their article (what does this mean?). If published, this will include your full peer review and any attached files.

Reviewer #1: **Yes: **Ana K. Rentsch

Reviewer #2: No

---

## [Author Response · Author response to Decision Letter 0]

19 Mar 2024

Editor:

Discussion of the results requires more depth in interpreting the results.

We have added new portions (L 628 - 640), removed certain sections (L 611 – 627, 642- 666) and rewritten (L 567 - 610) to bring more depth in interpreting results as well as make the discussion succinct. 

The statistical model needs clarification.

We used a combination of estimates, confidence intervals and p values to describe relationship with response and explanatory variables as described in L341 - 342., rather than use p < 0.05 as cut-off. We also wanted to try the approach of providing as many results as possible without the use of terms such as statistically significant and non-significant as advocated by Wasserstein et al., 2019, Hulbert et al., 2019, Berner and Amrhein (2022) etc. We followed a similar method as Rosenberger et al. (2022) to report our results. 

The entire manuscript needs a brief, more in-depth and clearer writing, with recent references cited.

We have removed and rephrased parts in all the sections to improve the clarity as well reduce the length of the manuscript. We have also added more references to support out claims. 

Reviewer #1: 

This study tested the association between spatial memory and learning (holeboard test) and voluntary ramp use in laying hen pullets. They found no within-bird consistency for ramp use behaviour. They did find relationships between ramp use frequency and the performance in the holeboard test, some of them supporting their predictions and some to the contrary. The experimental design is sound, though the statistical analyses are perhaps unnecessarily complicated. If no statistician was involved, I would recommend that the authors consult a statistician. If there was a statistician involved, then the section on statistics needs a bit of clarification. The manuscript is well written though it is lengthy and unnecessarily wordy and repetitious at times. My biggest concern is the discussion. There appears to be a lack of references and the structure can be improved by reducing redundant elaborations. Once cleaned up, this manuscript can be a valuable contribution to the laying hen literature.

Thanks for the detailed review and valuable feedback. After talking to a researcher with statistical expertise we have decided to remove the repeatability analysist because the sample size is on the lower end to calculate repeatability of ramp use. We have hence removed parts pertaining to repeatability analysis in all section of the manuscript. We also reanalysed the phase change analysis to include mean ramp use frequency (MRF) as a continuous variable. This analysis included all the birds unlike the previous analysis which used ramp use a categorical variable with high (ten birds with highest MRF) and low ramp users (ten birds with lowest MRF).

Major Comments

Stay consistent with the numbering (spelling out numbers under 10), please check the whole manuscript for inconsistencies.

Thanks. We have changed the numbering accordingly.

Abstract

You need to introduce terminology somewhere: chicks vs hens vs pullets. This journal is not poultry-specific.

We don’t think it is necessary. They are universally used terms and not specific to the paper and can be easily obtained if the reader does a quick web search. 

L27: what analyses? Give a model or analysis type to help understand the results.

We have changed it to Mixed-model analysis.

L30-33: These results are a bit confusing. First, they are trends/tendencies and not significant results, so you should state that. Second, it is unclear what the unit of those estimates are.

The estimates are back transformed, so it is integer scale (number of errors were positive integers including zero). The estimate here is a slope. For example, a 0.94 estimate means for one unit of increase in ramp use there is a 6% decrease in errors made. The change in errors between x and x+n ramp use would be n*0.6. 

For interpretation of statistics, we did not use p < 0.05 as cut-off. Rather we used a combination of estimates, confidence intervals and p values to describe relationship with variables as described in L341 - 342. We also wanted to try the approach of providing as many results as possible without the use of terms such as statistically significant and non-significant as advocated by Wasserstein et al., 2019, Hulbert et al., 2019 etc., 

L37: Please state why this would be relevant/ interesting.

With the given word limit, it is unfortunately not possible to have another sentence in the abstract, without losing information that we think are critical. Moreover, the relevance of the topic is covered in the introduction part.

Introduction

L63: I would suggest exchanging the Campbell reference with Widowski and Torrey’s book chapter on rearing young birds for adaptability: https://linkinghub.elsevier.com/retrieve/pii/B9780081009154000038. This statement refers to a shift in research interest based on the shift in management. Both the Ferreira and Widowski and Torrey references are revisions of the literature and the importance of research in this area. The Campbell paper is an experiment that can be mentioned later when discussing specific findings.

Thank you for the suggestion. We have changed it accordingly.

L108-116: This has to go before the research question/objectives (L106-107). Then, add lines 106-107 to the beginning of the next paragraph that contains your predictions.

Thanks for the feedback. We have rearranged the sentences and rephrased the research objective (L125 - 128)

Methods

L130-131: It’s a bit confusing to point to your other paper for the pen schematics. I immediately looked at your Figure 1 and expected to see a pen. I would suggest you provide schematics or a photo here. Alternatively, add the reference after your description and say that more information and schematics can be found in Johny et al. (28). But I would prefer a photo or schematics in this paper.

We have added the sentence you suggested in L 153. Thank you.

L157: How did you choose your random sample? How did you make sure there was no bias (e.g., first catches could be bold, last in the box could be most fearful).

Its not a random sample. Chicks were arbitrarily assigned to the pens. At the time of population, they came in crates. We picked the birds from the crates arbitrarily, applied colour marks and then introduced them to the pens. Since they were all constrained in small crates which did not allow any movement, we think there were no bias due to the catch order.

L169-173: Unclear what you mean here. What treatments? How were they managed? Why is this important?

The treatments were three different artificial cues that were aimed at improving ramp use and a control with no cue that were part of an additional experiment (Johny et al., 2023). The treatment groups influenced the variation in use of ramps, in addition to the individual-level differences. So, we added the treatment as control variable to account for the variation in ramp use due to the cues. We have added this information in L186 – 191. 

L292-295: This section needs clarification. I do not understand it. Do you mean the slopes in your statistical model? I also don’t understand why you group the birds into high and low ramp users. I though your sample was random. Did you have two clusters? Elaborate on this and on why you chose to create these categories instead of using their actual frequencies.

We changed the analysis and included MRF as a continuous variable like the rest of the analysis. This allowed us to use all the birds unlike the previous analysis which included ramp use as categorical variable with high (ten birds with highest MRF) and low ramp users (ten birds with lowest MRF).

L306-308: Why did you use bootstrapping to get the main effect p-values? To the best of my knowledge, this method is used to reduce a model and get p-values for reduced factors only. The main effects can be assessed as part of the model. Was a statistician involved in this analysis? If so, I will retract my comment. Otherwise, I would like you to consult a statistician.

We used the method suggested by PD Dr. Lorenz Gygax, HU Berlin. Although we did not consult him specifically for this experiment, one of the authors took a statistical course from him. We performed the analysis according to the framework provided in the course. Some major additions to the statistics from the course were:

1. Setting sum contrasts for the categorical variables and standardising continuous variables which provide interpretable main effects, even in the presence of interaction effects. This approach provides parameters such as p-values, estimates and confidence intervals of all main effects without the need to dropping interaction terms or reducing the global model that was initially chosen because of their biological relevance.

2. Using bootstrapping to obtain p-values, which are more conservative than likelihood ratio test, due to its robustness in dealing with influential points (Luke, 2016). 

L326: If you only include non-zero observations, are you not biasing the analysis? What if a bird uses it occasionally? As I understand it, you would overestimate the repeatability by only including the observations where the bird happened to use the ramps and excluding observations where it didn’t.

We removed the repeatability analysis after consulting a researcher with statistical expertise because our sample size is on the lower limits required for repeatability analysis. 

However, to answer your question, we did two different analyses to account for the huge number of zeroes. The first analysis is a binary model and had zeroes included in it. As it was binary, ramp use was coded as if a focal bird used ramps in a particular bout (1) or not (0). In the second analysis, we excluded the observations where an individual birds did not use the ramps. In that case, your argument is right. Taking out zeroes increases the variation in data and since repeatability is the ratio of individual level variance to total variance, one would expect an increase in repeatability. But as was mentioned in the discussion section that is now removed, the total variability of data is less, because of the 3 min observation window we used. That is likely a reason why we still didn’t see repeatability in ramp use, even after excluding zeroes.

Results

I recommend making the sentence about the effects of p < 0.01 less vague. If you decide to report tendencies (which you need to clarify in the stats section), report them without embellishments. E.g., ‘errors tended to decrease with trial number’ instead of ‘with very limited statistical support, Y made slightly fewer errors’.

As we wrote in response to your comment in the abstract part, we are using the effect size, uncertainty intervals and p-values to describe our results, rather than using the p-value cut-off of <0.05. So, we also opt to change the use of terms such as significant (p<0.05) and trend (0.05<p< ~0.08). We do understand that this is not the norm in the field. We used a similar method as in Rosenberger et. al, 2022, to describe our results. 

L385-386: I am still confused about where these high and low classifications come from… and now there is a medium group. Was I supposed to know that? Maybe I missed it.

We wanted to provide the estimated marginal means (EMM) for the number of errors made. We selected birds with low, median, and high MRF as a few examples to provide the EMMs, instead of providing EMMs of all MRF observations. Additionally, we removed the low and high ramp user category from the phase change analysis as mentioned earlier.

L448: I am not sure if removing these data points is appropriate. What if the group you claim learned faster actually had trials in there that were not completed? I recommend analysing it with the complete data set or declaring which ramp use groups the omitted data points came from if they were exclusively from low ramp users.

9, 1 and 7 out of 480, 960 and 370 observations from cued, uncued and reversal phases, respectively were omitted, which we think will have a negligible effect on the analysis. 

Discussion

There is a general lack of references supporting sweeping statements in this discussion. Please put your results into the context of existing literature and how your study adds to the knowledge base. There is also a lot of repetition that could be cut down (e.g., repeating of your own results in every paragraph).

Thanks for the feedback. We have restructured the discussion based on your comments.

L497-505: But your birds’ ramp use behaviour was inconsistent over time. How would that translate to personality, which is what you are implying but not spelling out here.

Our analysis did not reveal consistency in ramp use. But given results from Montalcini et al., 2023a, Montalcini et al., 2022, we would expect this behaviour to be repeatable. Also the movement patterns in aviaries across individuals have been shown to be consistent (Rufener et al., 2023, Montalcini et al., 2023a, Montalcini et al.,2023b). We think it’s a limitation of the method (three-minute observation period) as well the low sample size (which led us to remove the analysis on repeatability of ramp use) which rendered the ramp use behaviour non-repeatable in our analysis. 

The argument we make is that differences in ramp use (a movement pattern) and spatial cognition could be driven by individual differences in activity, exploration, and fearfulness. The differences in movement behaviours have been shown to correlate to differences in behaviours such as activity, exploration as mentioned in L533-538. On that note, it is possible that ramp use might also be driven by individual differences in activity (Rufener et al., 2023) and exploration (de Haas et al., 2017; Rodenburg et al., 2003) which has been shown to be repeatable in hens.

L568-572: Why would this only affect the uncued phase? If HR pullets do better in the cued but not the uncued phase, would this mean there is no difference in spatial learning (location) but only in association learning (colour)?

Yes, this is one possible explanation. We have added this in L627- 640. Thanks for the suggestion.

L580-583: WoA 19 is not the start of peak of lay, and WoA25 is not sexual maturity… it would be the reverse. Are you sure these ages and the terminology have the correct references? I also think that these are two different aspects. Sexual maturity would end the ontogenesis and be accompanied by hormonal changes and, in the case of laying hens, also with a change in housing. The peak of lay is well after that and is not initiated by hormonal or housing changes. I would be hesitant to group those two results and make one statement about behavioural consistency during ontogeny.

As we removed the ramp use repeatability analysis, this part of discussion is no longer in the manuscript.

L590-593: As ramp use was the behaviour in focus, why choose an observation period that you think is too short? If you had good reason to assume it was appropriate before you started, you should not backtrack now but rather accept your results. If you have reason to believe it was inappropriate based on new information after the fact, please add a section with limitations and explain your initial rationale and what changed.

This part is removed from the manuscript as it part of the repeatability analysis.

L593-596: No. You tested a hypothesis and did not reject the null hypothesis. If you have reason to assume a type 2 error, you must provide it. Otherwise, you cannot state your initial prediction after not confirming it. Also, what is ‘short-term’ consistency even? If you introduce a vague term like that, you have to define it and back it up with literature.

This part is removed from the manuscript as it part of the repeatability analysis.

Minor Comments

Do not be alarmed by the number of comments, most are very small and should be easily addressed.

Abstract

L19: ‘affect the brain’ is very vague. Be more specific or don’t mention the brain.

Removed the brain part.

L25: ‘observation period’ what ages are those? Instead of observation period. Give an age range.

Added the observation days.

L25-26: ‘spatial cognition’ be more specific on what aspect of spatial cognition

---

## [Decision Letter · Decision Letter 1]

4 Apr 2024

Mind the ramp: Association between early life ramp use and spatial cognition in laying hen pullets.

PONE-D-23-28746R1

Dear Dr. Alex Johny

We’re pleased to inform you that your manuscript has been judged scientifically suitable for publication and will be formally accepted for publication once it meets all outstanding technical requirements.

Kind regards,

Lamiaa Mostafa Radwan, Ph.D.

Academic Editor

PLOS ONE

Additional Editor Comments (optional):

Accept

Reviewers' comments:

Reviewer's Responses to Questions

**Comments to the Author**

1. If the authors have adequately addressed your comments raised in a previous round of review and you feel that this manuscript is now acceptable for publication, you may indicate that here to bypass the “Comments to the Author” section, enter your conflict of interest statement in the “Confidential to Editor” section, and submit your "Accept" recommendation.

Reviewer #1: All comments have been addressed

Reviewer #2: All comments have been addressed

2. Is the manuscript technically sound, and do the data support the conclusions?

Reviewer #1: Yes

Reviewer #2: Yes

3. Has the statistical analysis been performed appropriately and rigorously? 

Reviewer #1: Yes

Reviewer #2: Yes

4. Have the authors made all data underlying the findings in their manuscript fully available?

Reviewer #1: Yes

Reviewer #2: Yes

5. Is the manuscript presented in an intelligible fashion and written in standard English?

Reviewer #1: Yes

Reviewer #2: Yes

6. Review Comments to the Author

Reviewer #1: As far as I can tell, the authors have addressed all my concerns and greatly improved the quality of this manuscript. Well done

Reviewer #2: Its a well written paper, it would be great to go ahead with this topic and evaluate the impact of ramps with bird's production performance and various welfare parameters.

7. PLOS authors have the option to publish the peer review history of their article (what does this mean?). If published, this will include your full peer review and any attached files.

Reviewer #1: **Yes: **Ana K. Rentsch

Reviewer #2: No
